# Histological evaluation of the distribution of systemic AA-amyloidosis in nine domestic shorthair cats

**Valentina Moccia**[1]*, **Anne-Cathrine Vogt**[2,3], **Stefano Ricagno**[4], **Carolina Callegari**[5], **Monique Vogel**[2], **Eric Zini**[6,7,8], **Silvia Ferro**[1]

**1** Department of Comparative Biomedicine and Food Science, University of Padua, Legnaro, PD, Italy, **2** Department for BioMedical Research, Faculty of Medicine, University of Bern, Bern, Switzerland, **3** Graduate School for Cellular and Biomedical Sciences, University of Bern, Bern, Switzerland, **4** Department of Biosciences, University of Milano, Milano, MI, Italy, **5** IDEXX Laboratories, Milano, MI, Italy, **6** Department of Animal Medicine, Production and Health, University of Padova, Legnaro, PD, Italy, **7** AniCura Istituto Veterinario Novara, Granozzo con Monticello, NO, Italy, **8** Clinic for Small Animal Internal Medicine, Vetsuisse Faculty, University of Zurich, Zurich, Switzerland

* valentina.moccia@phd.unipd.it

**Data Availability Statement:** All relevant data are within the paper and its Supporting information files.

## Abstract

Amyloidosis is a group of protein-misfolding disorders characterized by the accumulation of amyloid in organs, both in humans and animals. AA-amyloidosis is considered a reactive type of amyloidosis and in humans is characterized by the deposition of AA-amyloid fibrils in one or more organs. In domestic shorthair cats, AA-amyloidosis was recently reported to be frequent in shelters. To better characterize this pathology, we report the distribution of amyloid deposits and associated histological lesions in the organs of shelter cats with systemic AA-amyloidosis. AA-amyloid deposits were identified with Congo Red staining and immunofluorescence. AA-amyloid deposits were then described and scored, and associated histological lesions were reported. Based on Congo Red staining and immunofluorescence nine shelter cats presented systemic AA-amyloidosis. The kidney (9/9), the spleen (8/8), the adrenal glands (8/8), the small intestine (7/7) and the liver (8/9) were the organs most involved by amyloid deposits, with multifocal to diffuse and from moderate to severe deposits, both in the organ parenchyma and/or in the vascular compartment. The lung (2/9) and the skin (1/8) were the least frequently involved organs and deposits were mainly focal to multifocal, mild, vascular and perivascular. Interestingly, among the organs with fibril deposition, the stomach (7/9), the gallbladder (6/6), the urinary bladder (3/9), and the heart (6/7) were reported for the first time in cats. All eye, brain and skeletal muscle samples had no amyloid deposits. An inflammatory condition was identified in 8/9 cats, with chronic enteritis and chronic nephritis being the most common. Except for secondary cell compression, other lesions were not associated to amyloid deposits. To conclude, this study gives new insights into the distribution of AA-amyloid deposits in cats. A concurrent chronic inflammation was present in almost all cases, possibly suggesting a relationship with AA-amyloidosis.

**Funding:** The authors received no specific funding for this work.

**Competing interests:** The authors have declared that no competing interests exist.

## Introduction

Amyloidosis refers to a group of protein-misfolding disorders characterised by the accumulation of amyloid deposits in organs. In humans and animals, amyloid is composed of proteinaceous insoluble fibrils, which result from a conformational change into beta-sheet structure of proteins that are soluble in physiological conditions [1–3]. Amyloidosis can be classified into two major categories, localized and systemic. In localized amyloidosis, amyloid fibrils accumulate at just one site, while in the systemic form, they accumulate in multiple organs [1, 3]. AA-amyloidosis is a type of amyloidosis caused by the deposition of AA-amyloid fibrils in various organs [4]. It is a multifactorial disease that affects humans, domestic, laboratory and wild animals [1, 4]. The precursor protein of AA-amyloid fibrils is serum amyloid A (SAA), which is an acute phase protein. The disease has often been associated with long term inflammatory stimuli, but the structure of SAA, increased synthesis, misfolding tendency, and the presence of an acute phase response are all variables that may promote the development of the disease [2–4]. However, the pathogenesis of AA-amyloidosis has not been completely understood thus far. Studies in mice and minks have shown that it can be triggered by the intravenous administration of amyloid-enhancing factor or amyloid fibrils, and disease spread through oral transmission has been experimentally demonstrated in cheetahs and chickens [1, 5, 6]. Among felids, AA-amyloidosis has been reported not only in cheetahs but also in cats, Siberian tigers, and black-footed cats [7–13]. In captive cheetahs, AA-amyloidosis is an important cause of morbidity and mortality, with a prevalence up to 70 per cent in animals at necropsy [6, 9, 14]. Considering the high prevalence of the disease, often associated with inflammatory processes, and the identification of amyloid fibrils in the feces, the oral route is considered a potential mode of transmission in this species. Consequently, these findings have led to increased attention to environmental and breeding conditions to prevent the spread of the disease among captive cheetahs [6, 9]. In cats, the most commonly reported forms of amyloidosis are the amylin-derived amyloidosis of the pancreatic islets in older animals and the systemic AA-amyloidosis in Siamese and Abyssinian breeds, which seem to have a familial predisposition [11, 15–18]. In contrast, in domestic shorthair cats, systemic AA-amyloidosis is rare and poorly characterized [10–13]. In one study, an association between systemic AA-amyloidosis and experimental infection with feline immunodeficiency virus (FIV) was documented in domestic shorthair cats, while other predisposing factors have not been described [10, 16, 19]. Interestingly, a recent study investigating the prevalence of AA-amyloidosis in three cat shelters in Northern Italy showed an unexpectedly high prevalence of the disease, with a total of 48/79 positive cats. Moreover, the authors found SAA presence in the bile of affected cats, suggesting a possible fecal-oral route of transmission [20]. Additionally, the cryo-electron microscopy structure of AA-amyloid fibrils derived from one shelter cat has been recently reported [21]. However, a thorough histopathological assessment of AA-amyloid distribution and associated lesions in domestic shorthair cats is lacking. Therefore, considering the limited information on the histopathological features of AA-amyloidosis in domestic shorthair cats and the recent findings suggesting an underestimation of this disease, this study aims to characterize systemic AA-amyloidosis, focusing on the histopathological description and scoring of amyloid organ distribution and on the reporting of the concurrent histological lesions.

## Materials and methods

### Animals and clinical data

Cats that spontaneously died or were euthanised for the severity of their clinical condition between January 2019 and February 2020 coming from 3 shelters located in the Piemonte

region (Northern Italy), were prospectively enrolled when multiorgan sampling was performed within 6 hours from death. Each shelter was approximately 100 km far away from each other.

## Tissue samples and histology

In each cat, tissue sampling was performed after external examination and opening of the abdoment and the thorax. Sampling was conducted from both normal tissues and lesions, if present. At least 11 organs were samples from each cat and the liver, a kidney, a parotid, the stomach, the large intestine, a lung, the urinary bladder, a lymph node and one eye were always included Tissues were fixed in 10 per cent buffered formalin for at least 24 hours and then embedded in paraffin. Four to 5 μm thick tissue sections were stained with Congo red (CR) with and without potassium permanganate pre-treatment. Briefly, sections were stained manually in hematoxylin for 10 min, then stained with CR (PanReac) for 50 min and dehydrated quickly in 100per cent alcohol. In the slides pre-treated with potassium permanganate, steps were as follows: sections were oxidized in potassium permanganate for 3 minutes and then neutralized with 5 per cent oxalic acid solution. Sections were then stained in hematoxylin for 10 minutes, followed by CR for 50 minutes and dehydrated in 100 per cent alcohol. All sections were cleaned in xylene and then mounted. CR staining was examined under standard and polarized light microscopy (Olympus BX41) to detect the green birefringence associated with amyloid deposits and to describe their distribution. Only cats with more than 2 CR-positive organs were included in the study. A score from 0 to 2 was applied to the CR-positive tissue sections based on the amount and distribution of green birefringent material after examination under polarised light microscopy. Zero was assigned to sections without CR-positive material, 1 to sections with mild focal CR-positive deposits, and 2 to sections with moderate or severe, focal or multifocal CR-positive deposits. Pre-treatment with potassium permanganate was performed to characterize the amyloid fibrils; the lack of birefringence after treatment with potassium permanganate was considered compatible with AA-amyloidosis [22]. To describe histological lesions, additional tissue sections of amyloid-positive cats were stained with hematoxylin and eosin (HE) using an automatic stainer (AutoStainer XL, Leica Biosystems), and examined under standard light microscopy.

## Immunofluorescence (IF)

To confirm that CR-positive tissues were consistent with AA-amyloid deposits, at least one organ per cat was evaluated by IF. Prior to all staining, tissues were deparaffinized (xylol, 100 per cent (v/v), ethanol, 100 per cent (v/v), ethanol 95 per cent (v/v) ethanol, 70 per cent (v/v) ethanol, 50 per cent (v/v) ethanol, and finally ddH2O). To typify amyloid, sera were obtained from BALB/cOlaHsd mice immunized with peptides conjugated with virus-like particles (VLPs) [21]. Previously described residues from feline serum amyloid A, 41–50 (MREANYIGAD), 63–74 (QRGPGGAWAAKV) and 109–122 (EWGRSGKDPNHFRP) of Uniprot entry P19707, were selected as antigens and coupled to VLPs [12]. Mice were subcutaneously injected with 30ug of the different vaccines and boosted at day 14. Blood was collected at day 21 and IgG specific for the three peptides was measured by ELISA. For the immunofluorescence staining, mouse immune sera was diluted 1:10 in a self-made blocking buffer (BSA 2 per cent, Triton-X-100 0,5 per cent in PBS) and tissue sections were incubated overnight at 4˚C. As secondary antibodies, a goat anti-mouse monoclonal [AB27942969] IgG conjugated to biotin (ab1030–08, Southern Biotech) followed by streptavidin conjugated Alexa546 (s11225, Molecular Probes) was used. The 1 per cent thioflavin S (T1892, Sigma-Aldrich) in ddH2O was used to stain amyloid aggregates. In the presence of biotinylated antibodies an avidin/

biotin blocking kit (Vector Laboratories) was used to block endogenous biotin according to the manufacturer. All pictures were acquired with an Axio Imager. A2 and a Carl Zeiss AxioCam.

## Results

Starting from a group of 20 cats, 11 cats were excluded because 10 were CR-negative and one had only one positive organ (i.e., the spleen), hence the diagnosis of systemic amyloidosis was not confirmed. Nine cats with amyloid deposits in multiple organs and with the AA-amyloid-type confirmed at CR pre-treated with potassium permanganate and IF were then included in the study. The number of CR-positive organs and their scores are shown in Table 1.

149 organs from the nine cats included in the study were available, and 86 of these organs were positive at CR under polarized light. All of the kidneys (9/9), spleens (8/8), adrenal glands (8/8), small intestines (7/7), gallbladders (6/6) and tongues (3/3) and most of the livers (8/9), stomachs (7/9), large intestines (7/9), and hearts (6/7) resulted positive. Other positive organs were parotids (3/9), pancreases (5/8), lungs (2/9), urinary bladders (3/9), lymph nodes (4/9), and skin (1/9). Conversely, none of the eyes (0/9), skeletal muscles (0/7) and brains (0/7) had amyloid deposits (Table 1). Amyloid deposits with a score 2 were observed in 71 of 87 (87.6 per cent), whereas the remaining 16 (18.4 per cent) organs scored 1. Among the organs with score 1, 3/7 were stomachs, 2/4 lymph nodes, 1/2 lungs, and 3/6 hearts. Among the 8 skin samples, the only positive one had a score equal to 1. In organs with amyloidosis, deposits were observed as follows. In all the three positive parotids Fig 1a, deposits were multifocal or diffuse in the interstitium. In one of them, deposits were also periductal and vascular. Besides amyloid deposits, parotids did not present any other lesions. In all the three tongues Fig 1b deposits were multifocal and subepithelial in the propria-submucosa; two of the three tongues

**Table 1. List of negative and positive organs with amyloid-scoring.**

| Organ | Score 0 | Score 1 | Score 2 | Positive/total organs |
|---|---|---|---|---|
| Parotid | 6 | 0 | 3 | 3/9 |
| Tongue | 0 | 0 | 3 | 3/3 |
| Stomach | 2 | 3 | 4 | 7/9 |
| Smallintestine | 2 | 0 | 7 | 7/7 |
| Largeintestine | 2 | 0 | 7 | 7/9 |
| Liver | 1 | 1 | 7 | 8/9 |
| Gallbladder | 0 | 1 | 5 | 6/6 |
| Pancreas | 3 | 1 | 4 | 5/8 |
| Lung | 7 | 1 | 1 | 2/9 |
| Kidney | 0 | 1 | 8 | 9/9 |
| Urinarybladder | 6 | 1 | 2 | 3/9 |
| Adrenalgland | 0 | 1 | 7 | 8/9 |
| Spleen | 0 | 0 | 8 | 8/8 |
| Lymphnode | 5 | 2 | 2 | 4/9 |
| Heart | 1 | 3 | 3 | 6/7 |
| Eye | 9 | 0 | 0 | 0/9 |
| Brain | 7 | 0 | 0 | 0/7 |
| Skin | 7 | 1* | 0 | 1/8 |
| Skeletalmuscle | 7 | 0 | 0 | 0/7 |

* Amyloid deposits were associated to the skin and the mammary gland tissue included in the sample.

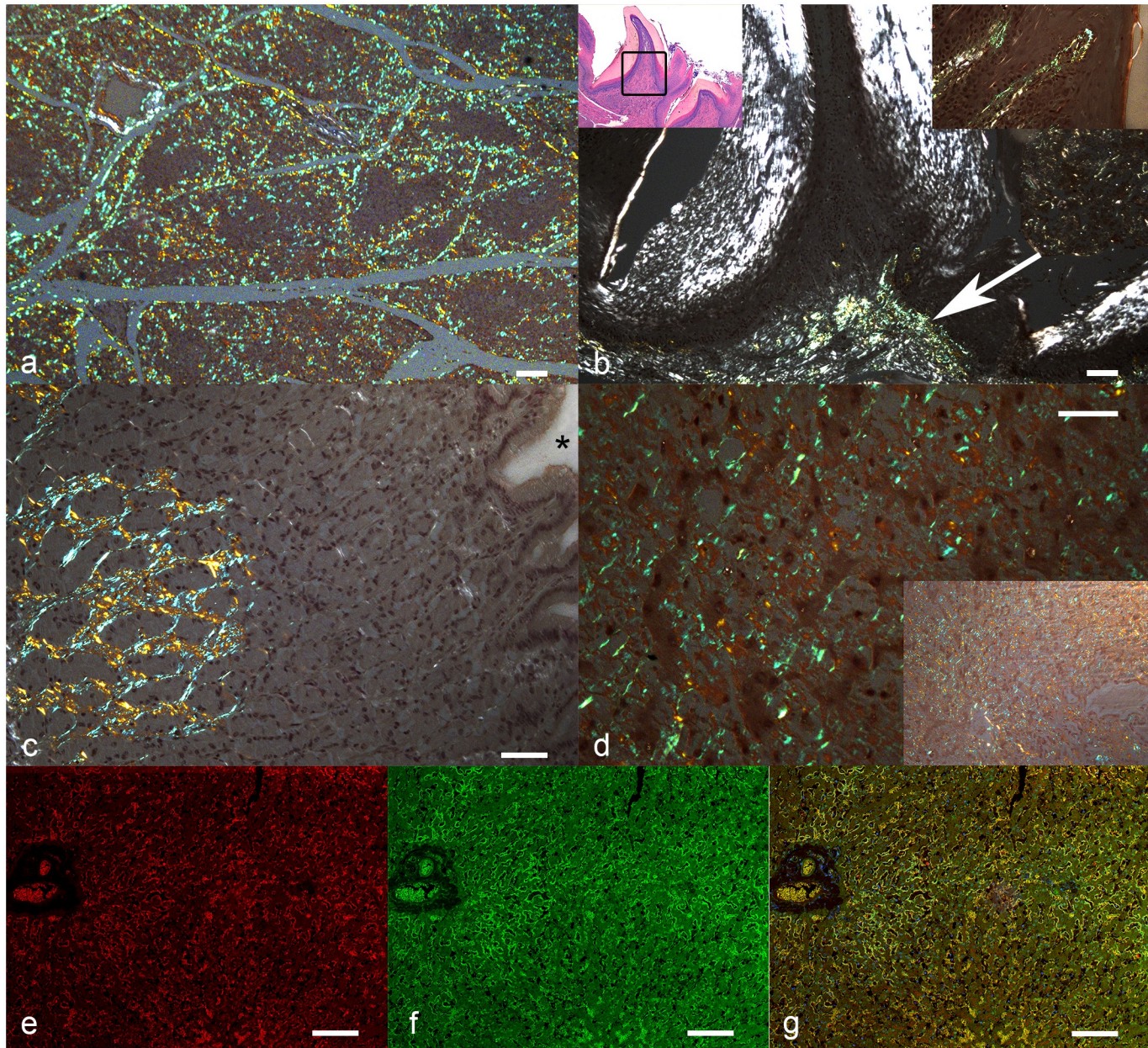

**Fig 1. Systemic AA-amyloidosis, cat.** a-d) Congo red. Amyloid deposits are highlighted by apple-green birefringence of the Congo red positive material under polarized light. a) Parotid. Moderate disseminated deposits. The figure shows that the lobular gland structure is preserved. Bar = 100 um. b) Tongue. Moderate deposit in the superficial lamina propria under a papilla (arrow). Left inset: lower magnification showing the superficial mucosa of the tongue with some papillae. The black square represents the area of the main image. Haematoxylin and eosin. Right inset: higher magnification showing subepithelial amyloid deposits. Bar = 80 um. c) Stomach. Abundant interstitial amyloid in the submucosa. Asterisk: lumen of the stomach. Bar = 50 um. d) Liver. Diffuse and severe amyloid deposits in the Disse space along the hepatocytes. Bar = 50 um. Inset: lower magnification, all the parenchyma is involved. e-g) Liver. Fluorescence-immunohistochemistry. Bar: 50 um. e) Interaction of amyloid deposits with serum (red). f) Amyloid deposits stained with ThioflavinS (green). g) Interaction of amyloid deposits with serum (yellow for colocalization) and nuclei stained with DAPI (blue).

presented concurrent lesions including a focal subacute ulcerative glossitis in one tongue and eosinophilic granuloma in the other S1 Table.

In the seven stomachs Fig 1c, deposits were in the lamina propria, focal in three cases and diffuse in the remaining four. Other concurrent lesions were not observed. In the 14 positive

intestines (of which seven were small and seven were large intestines) deposits were distributed multifocally or focally extensive along the lamina propria. Of these affected intestines, in one small intestine and in two large intestines, deposits extended to the submucosa; in one small intestine amyloid was also present in the vessel walls, and in one large intestine, deposits were evident in vessel walls and in the muscularis mucosae. In all the 14 samples, a concomitant mild to moderate chronic enteritis was observed. Additionally, in one large intestine, enteritis was associated with fibrinous peritonitis due to feline infectious peritonitis S1 Table. Furthermore, two small and two large intestines presented deposits without other concurrent lesions, and two large intestines without deposits had mild to moderate chronic enteritis. In the positive eight livers Fig 1d–1g, deposits were observed in the Disse space and diffusely in the interstitium among the hepatocytes, often leading to hepatocellular atrophy. In four livers, deposits were also vascular. In two cases, the Glisson's capsule was also involved by amyloid deposits. Four of the eight livers had a wide range of concurrent lesions, including one nonspecific periportal hepatitis associated with mild lymphocytic cholangitis, one focal biliary cysts, one with fibrinous-hemorrhagic perihepatitis and polycystic liver disease, and one case of single-cell necrosis associated with hepatocellular disarchitecture and pigment accumulation S1 Table. In the gallbladder Fig 2a, amyloid deposits were observed in the mucosa of three samples, in the vascular walls in two samples, and in both of them in one case. Notably, in one case, positive CR material was also documented in the bile contained in the lumen of the gallbladder. We observed other lesions only in one gallbladder, which presented a mild diffuse lymphoplasmacytic cholecystitis S1 Table.

In the five positive pancreases, amyloid was mainly multifocal in the interstitium, in the islets, and in vascular and ductal walls. Two pancreases had concurrent mild pancreatitis, and one of them presented mild nodular hyperplasia S1 Table. The remaining two positive pancreases did not have other lesions, and three pancreases without deposits presented mild nodular hyperplasia. Considering the two positive lungs Fig 2b, one of them presented severe amyloid deposits which affected the vascular walls, the alveolar and peribronchiolar interstitium, and were evident also in the lumen of small vessels. In the second sample, deposits were mild and limited to one vessel wall. Both lungs presented moderate atelectasis and emphysema, which were associated with mild oedema in the second lung S1 Table. The seven negative lungs, although not affected by amyloid deposits, still presented a variety of other lesions, including two moderate subacute pneumonia with atelectasis; one chronic interstitial pneumonia associated with atelectasis and emphysema; one severe atelectasis and pulmonary artery hypertension; one focally extensive haemorrhage with emphysema and oedema, and one diffuse severe necrotizing bronchopneumonia. In the nine kidneys Fig 2c–2g, all presenting amyloid deposits, fibrils were segmentally or globally localized in the glomerulus, in or around the Bowman capsule, peritubular, perivascular, in vascular walls and, in general, diffuse in the interstitium of the cortex and medulla. Notably, one kidney had a mildly ulcerated pelvis resulting in the leakage of amyloidotic content into the renal pelvis Fig 2e and 2f. Seven kidneys concurrently presented chronic interstitial nephritis, one a severe cystic tubular ectasia, and one had a severe tubular atrophy and arterial hypertension S1 Table. In the three CR-positive urinary bladders Fig 2h, amyloid deposits were mainly mild and subepithelial, with one case where deposits reached the submucosa. One of the urinary bladders presented mild chronic cystitis with mild subepithelial oedema S1 Table. In the eight adrenal glands Fig 3a, all with amyloid deposits, fibrils were distributed in the cortices, medulla, vessels as well as in the capsules. Additionally, four adrenal glands had diffuse hyperplasia, while the other four did not present other lesions.

In the eight spleens Fig 3b, all with amyloid deposits, fibrils were observed in the trabeculae, parenchyma, vascular walls as well as in the capsules. Two of the specimens presented depletion of both red and white pulp, one each had mild to moderate follicular hyperplasia and mild

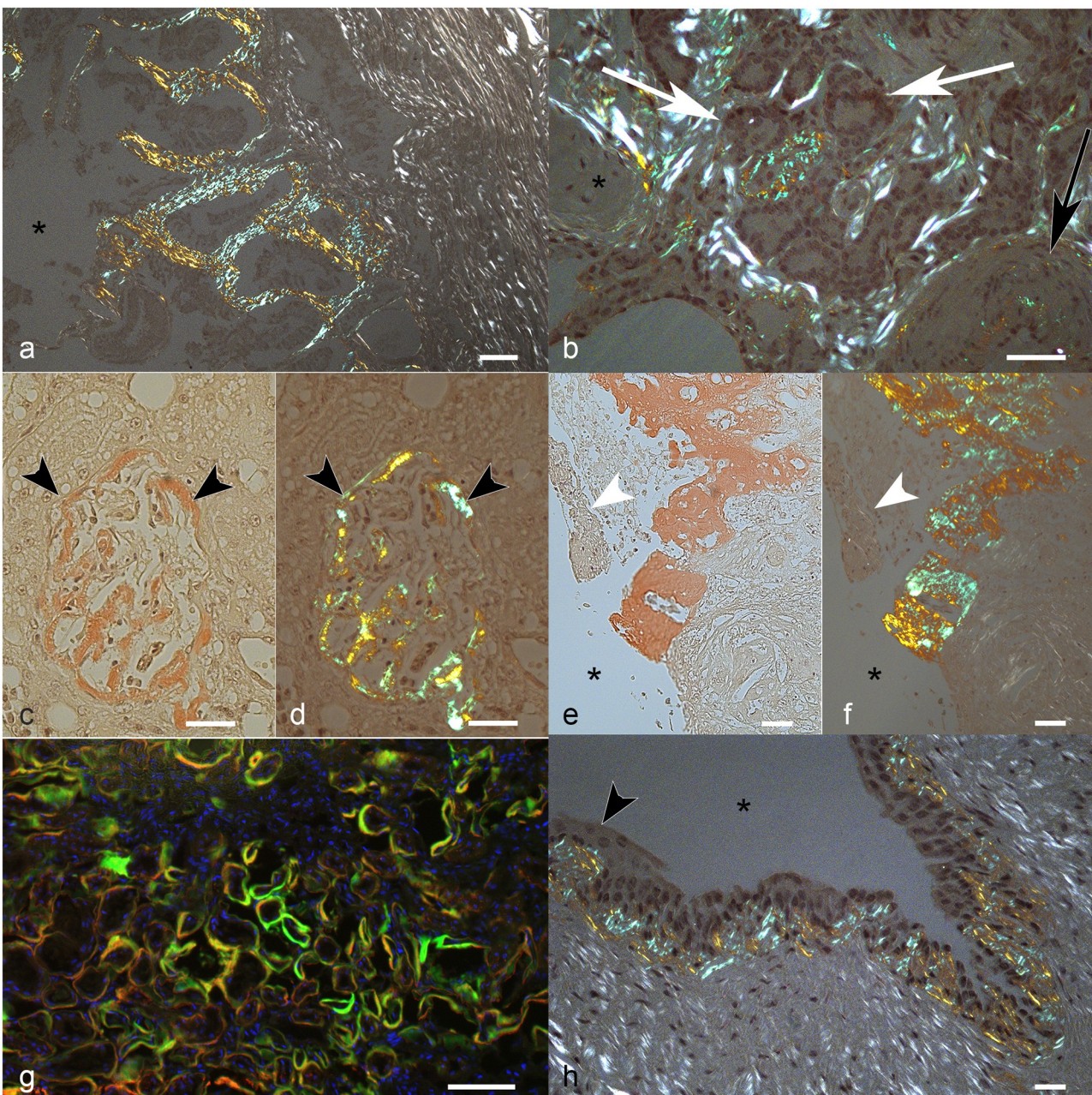

**Fig 2. Systemic AA-amyloidosis, cat.** Bar = 50 um. a) Gall bladder. Diffuse apple-green birefringent deposits in the superficial lamina propria. Asterisk: lumen. Congo red (CR). b) Lung. Apple-green birefringent amyloid deposits are evident around the bronchial glands (white arrows) and in the wall of a medium size artery (black arrow). Asterisk: peribronchial cartilage. CR. c-g) Kidney. c, d) Global distribution of amyloid in a glomerulus (between arrowheads). The Bowman's capsule is also involved. c) Amyloid stains red at light microscopy. D) Amyloid is birefringent and stains apple-green under polarized light. e, f) Renal pelvis. Abundant interstitial amyloid deposit in the medulla under the damaged (arrowhead) and ulcerated epithelium, apparently leaking to the lumen (asterisk). e) Amyloid stains red at light microscopy. f) Amyloid is birefringent and stains apple-green under polarized light. g) Fluorescence-immunohistochemistry of diffuse peritubular amyloid deposits. Colocalization (yellow) of interaction of amyloid deposits with serum (red) and amyloid deposits stained with ThioflavinS (green). Nuclei are stained with DAPI (blue). h) Urinary bladder. Diffuse apple-green birefringent subepithelial amyloid deposits. Asterisk: lumen; arrowhead: epithelium. CR.

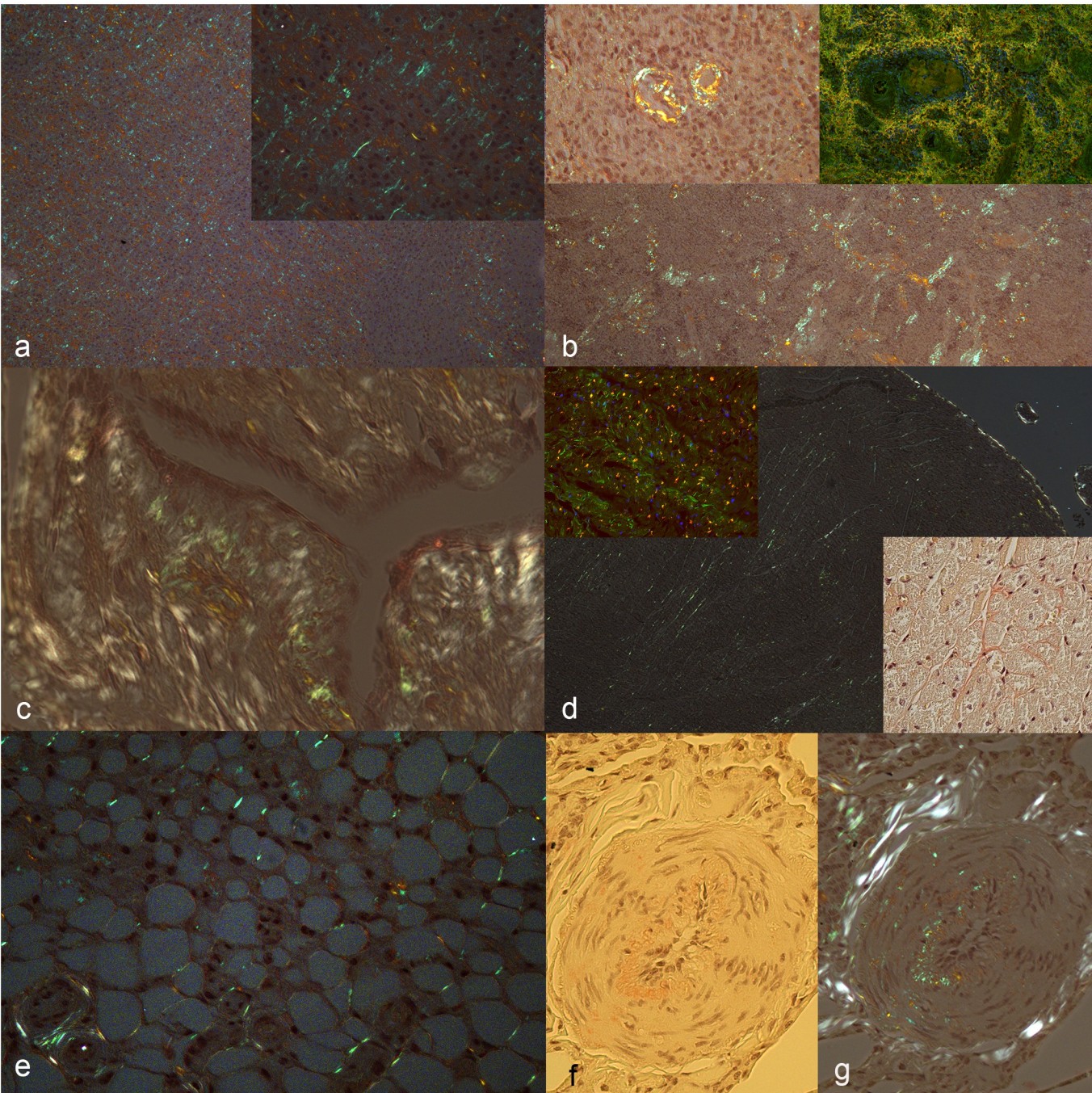

**Fig 3. Systemic AA-amyloidosis, cat.** Congo red (CR). Amyloid deposits are highlighted by apple-green birefringence of the CR-positive material under polarized light. a) Adrenal gland. Diffuse severe interstitial deposits. Bar = 50 um. Inset: higher magnification. b) Spleen. Multifocal interstitial moderate amyloid deposits. Bar = 150 um. Upper left inset: vascular deposits. Upper right inset: fluorescence-immunohistochemistry. Colocalization (yellow) of the interaction of the diffuse amyloid deposits with serum and ThioflavinS. c) Mammary gland. Amyloid deposits in the subepithelial interstitium along a mammary duct. Asterisk: lumen. Bar = 50 um. d) Heart. Moderate multifocal interstitial deposits in the cardiac muscle. Asterisk: ventricular cavity; Arrowhead: endocardium. Bar = 100 um. Upper inset: fluorescence-immunohistochemistry. Colocalization (yellow) of the interaction of amyloid deposits with serum (red) and amyloid deposits stained with ThioflavinS (green). Lower inset: interstitial amyloid deposits stain red at light microscopy. e) Adipose tissue. Frequent multifocal amyloid deposits between the adipocytes. Bar = 50 um. f, g) Artery. Moderate deposits in the vessel wall (arrowhead) and in the interstitium at the periphery. f) Amyloid stains red at light microscopy. g) Polarized light. Bar = 50 um.

leukocytoclastic splenitis S1 Table. The remaining four spleens presented amyloidosis without other concurrent lesions. In the four positive lymph nodes, amyloid was in follicles and in germinative centres, perifollicular, perivascular, and in vascular walls. Among the four lymph nodes with amyloid deposits, one case had concurrent reactive lymphadenitis and one presented chronic fibrinous lymphadenitis S1 Table. In the two remaining cases, lymph nodes did not have other lesions. Among all the nine lymph nodes, one lymph node negative for amyloidosis presented reactive lymphadenopathy. In the only CR-positive skin sample out of eight, amyloid was multifocal and perivascular in the subcutis, in the hair follicle, and multifocally around the mammary ducts of a neutered female Fig 3c. We did not find any histological lesions in any of the available skin samples. In the six hearts Fig 3d with amyloid deposits, amyloid amount was mild in three and moderate to severe in the others. The distribution of fibrils was interstitial, perivascular, and vascular (arterial walls) in all organs, and in one also perineural. Hearts did not have any concurrent histological lesions. None of the seven skeletal muscle samples had amyloid deposits and any concurrent histological lesions. In all cases, eye and brain samples were negative for CR. In one case, the eye presented diffuse moderate chronic blepharitis with degeneration of the crystalline and anterior synechia of the iris. The brain of one cat had moderate, subacute, lymphoplasmacytic meningitis, and another cat had mild oligodendrogliosis with mineralization of choroid plexus vessel walls. In addition to the listed sites of amyloid deposits, we observed amyloid fibrils in the adipose tissues Fig 3e surrounding a CR-positive subcutaneous lymph node, a CR-positive small intestinal loop, and a CR-positive adrenal gland. Moreover, despite not considering the vascular compartment as an apparatus, it is noteworthy that vascular amyloid deposits were present in 6/9 positive cats Fig 3f and 3g. Among the nine cats with systemic amyloidosis, five presented concurrent moderate or severe chronic inflammation affecting more organs, three had mild chronic inflammation affecting the kidney and/or the intestine, and one had no detectable inflammation in any of the available organs. Permanganate pre-treatment CR resulted negative in all cases, suggesting that amyloidosis was of the AA-type, as also confirmed by IF in all the studied cats Figs 1–3 and S1 and S2 Figs.

## Discussion

AA-amyloidosis in domestic shorthair cats has always been reported as a rare disorder, although a recent study found an unexpected high prevalence of the disease in some Italian cat shelters [10–13, 16, 17, 20, 23, 24]. However, in all the reported cases of feline systemic AA-amyloidosis, a detailed histological description and the investigation of a wider subset of organs are lacking. In the present series, the organs most involved by amyloid deposits, both in terms of frequency and quantity (i.e. scoring) were the kidney, spleen, liver, intestine, and adrenal glands. Liver, spleen, and kidney are considered primary sites of AA-amyloid depositions both in human and animals, including cats [1, 4, 10, 12, 13]. The adrenal glands and intestines are also common deposition sites in felids, with the former being reported in domestic shorthair cats, cheetahs, and black-footed cats, and the latter in cats, Siberian tigers and black-footed cats [7–10, 12]. The lung and the skin (with an adjacent positive mammary gland in the only positive case) were the least involved among the affected organs of this study. Among the reports on feline systemic AA-amyloidosis, the lung is reported as a deposit site in domestic shorthair cats only in one case [11]. Hence, considering that we found only two positive lungs, one of which presenting mild deposits, our findings suggest that pulmonary deposition of AA-fibrils is uncommon in cats. Of note, lung involvement in systemic AA-amyloidosis is rarely reported in other species as well, among which the Siberian tiger, the bovine, the Japanese quail, the pekin duck, and the flamingo [7, 25–28]. To our knowledge,

skin and mammary gland involvement has never been investigated in cats with systemic AA-amyloidosis. However, these sites have been described in the cattle, where AA-amyloid deposits were mainly in the blood vessels, with a lesser involvement of the interstitium, similarly to our reported mammary gland positive sample [25]. Until now, stomach, gallbladder, urinary bladder, and heart, despite being investigated in other studies on feline systemic AA-amyloidosis, had never been reported as positive. While the stomach, the urinary bladder and the heart have been actually described in other felids, gallbladder deposits are reported only in humans and are still considered uncommon [7, 8, 29–31]. Considering that in our case series, the gallbladder resulted positive in all the examined samples (6/6), we might explain this high number of positive gallbladders despite the absence of previous reports, by the rare investigation of this organ [10, 12, 13]. Interestingly, we found multifocal or focal abundant amyloid deposits also in 1/3 of the parotid glands and in all the examined tongues. Although AA-amyloid deposits have already been described in salivary glands and in one tongue in cats, as for the gallbladder, these organs are infrequently sampled, and their involvement might be underestimated [10, 12, 32, 33]. Some organs, including the eye, brain and skeletal muscle, were spared from amyloid deposits in this study. In humans, ocular involvement has been described only in AL-amyloidosis and there are no reports of any type of fibrils in animals [34]. Brain involvement is also very rare in humans and animals; mainly vascular deposits have been reported in the central and peripheral nervous system only in flamingos [26]. The rare involvement of the eye and brain might be explained by their peculiar blood barriers which might have a role in protecting these organs by AA-amyloid spreading [35, 36]. Although skeletal muscles are reported more commonly in chickens and quails, they are not common AA-amyloid deposit sites in mammals [25, 28, 37] Interestingly, in chickens, amyloid deposits in the pectoral muscles were found to be correlated to vaccination and chronic inflammation [37]. Through the histological evaluation of these nine cases, we did not identify any specific lesion associated with amyloid deposits. The only exception was cell compression, invariably present, which can cause cell degeneration/necrosis followed by organ dysfunction in the most severe cases [4]. Hence, despite kidneys and intestines frequently presented amyloid and concurrent chronic nephritis and enteritis, respectively, a cause-and-effect relationship with inflammation was not detected. Likewise, also in humans no association with the presence of inflammatory lesions in the site of deposition has been described [38]. Although inflammatory reactions are not usually associated with amyloid deposits histologically, AA-amyloidosis is considered a consequence of chronic inflammatory diseases in the human literature since SAA is secreted during inflammatory conditions [39]. Unfortunately, we have no clinical information on the cats of the study to make hypotheses on the onset of the disease and its relevance. However, we histologically diagnosed a systemic mild or moderate inflammatory status in almost all (8/9) subjects. Moreover, infectious diseases are frequent in crowded and stressful environments, such as cat shelters [40, 41]. From the few collected information, we know that four of the eight cats with inflammation were Feline Leukaemia Virus (FeLV) positive, and one of them was also FIV positive. Only one of the studied cats did not have signs of inflammation in the examined organs. Interestingly, however, this cat had a long stay in the cat shelter of about 36 months, but other information on his health status is lacking. Experimentally, it has been demonstrated that AA-amyloidosis can be transmissible through oral and intravenous administration in bovine, avian, mouse, cheetah, and flamingo (1,39). However, the mechanisms of absorption and distribution are unknown. Researchers hypothesise that amyloid may spread like prions, through intestinal trans-epithelial transportation, through blood in monocytes or in exosomes [39, 42]. The potential spread of amyloid through circulation might explain why some of the examined organs were mainly affected in the vascular compartment while the parenchyma was involved to a lesser extent (e.g. in the lung). Moreover, since feline amyloid is very similar to cheetahs'

and amyloid fibrils were found in cheetahs' feces, we cannot exclude amyloid diffusion within tissues (e.g., the intestine) through the transit of fibrils within bowel content, which might also explain the frequent deposits in the upper layers of organs such as the intestine [6, 21]. Although the presence of amyloid fibrils has not be confirmed in cat feces, SAA has been recently detected in the bile of shelter cats positive for AA-amyloidosis, suggesting a possible fecal spread of SAA in cats, as supposed for cheetahs [6, 20]. The major limitations of this study are related to the heterogeneity of the available organs and the lack of available medical records. However, these limits do not invalidate the main aim of this study, which was the microscopic description of amyloid deposits and distribution in systemic AA-amyloidosis and the possible presence of concurrent lesions in the same sites.

## Conclusion

In this series, we describe microscopically the distribution of amyloid deposits and the concurrent lesions in organs of shelter cats with systemic AA-amyloidosis. Among the apparatus the gastrointestinal tract and the urinary tract were always involved. Among the organs with deposits, the most frequently involved were the kidney, the spleen, the liver, the intestine and the adrenal glands. The lung and the skin were the least frequently involved organs and deposits were mainly confined to the vascular compartment. In addition, the eye, the brain and the skeletal muscles were found negative in all the studied cases. Moreover, AA-amyloid deposition in the stomach, in the gallbladder, in the urinary bladder and in the heart is reported for the first time in the domestic shorthair cats. Finally, although we did not detect histological lesions associated to AA-amyloid deposit except for cell compression, a chronic inflammatory status was detected in most of the animals, supporting the hypothesis that it could be one of the factors involved in the pathogenesis of the disease, as described in humans with AA-amyloidosis. Further studies investigating AA-amyloidosis prevalence, organ distribution and protein structure in cats from different shelters might be useful to better understand the pathogenesis of the disease.

## Supporting information

**S1 Fig. Immunofluorescent staining AA-amyloid in the heart of a cat.** Amyloidogenic amyloid A aggregates were stained with anti-SAA (a, in red) and ThioflavinS dye (b, green). A merge image was created to visualize co-localization (Merge, c, yellow). Nuclei were stained using DAPI and were coloured blue.
(DOCX)

**S2 Fig. Immunofluorescent staining AA-amyloid in the spleen of a cat.** Amyloidogenic amyloid A aggregates were stained with anti-SAA (a, in red) and ThioflavinS dye (b, green). A merge image was created to visualize co-localization (Merge, c, yellow). Nuclei were stained using DAPI and were coloured blue.
(DOCX)

**S1 Table. Summary of the detected lesions in AA-amyloid positive organs and of amyloid score.**
(DOCX)

## Acknowledgments

We would like to acknowledge Mr. Valter Fiore for technical support.

## Author Contributions

**Conceptualization:** Eric Zini, Silvia Ferro.

**Data curation:** Valentina Moccia.

**Investigation:** Valentina Moccia, Eric Zini, Silvia Ferro.

**Methodology:** Anne-Cathrine Vogt, Eric Zini, Silvia Ferro.

**Supervision:** Silvia Ferro.

**Writing – original draft:** Valentina Moccia, Anne-Cathrine Vogt.

**Writing – review & editing:** Valentina Moccia, Anne-Cathrine Vogt, Stefano Ricagno, Carolina Callegari, Monique Vogel, Eric Zini, Silvia Ferro.

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
