## [Decision Letter · Decision Letter 0]

18 Aug 2023

PONE-D-23-16620Histological evaluation of the distribution of systemic AA-amyloidosis in nine domestic shorthair catsPLOS ONE

Dear Dr. Moccia,

Thank you for submitting your manuscript to PLOS ONE. After careful consideration, we feel that it has merit but does not fully meet PLOS ONE’s publication criteria as it currently stands. Therefore, we invite you to submit a revised version of the manuscript that addresses the points raised during the review process.

We look forward to receiving your revised manuscript.

Kind regards,

Khaled Abd EL-Hamid Abd EL-Razik, Ph.D.

Academic Editor

PLOS ONE

Journal Requirements:

Additional Editor Comments:

The abstract:

The abstract summarizes the obtained findings, but the histopathological findings need more focusing.

Introduction:

The objective of this study can be written in a better way and more interesting.

The Authers not highlighting the lack of information about the histopathological findings that could be the objective of this study.

In general, the introduction has sufficient information about the nature of amyloidosis and its prevalence.

Material and Methods:

Materials and methods were written subtitled in a good way but why not talk about precautions taken during collecting testes samples?

Consider providing more details on specific staining techniques, reagents and equipment used.

Results:

The results are presented clearly and concisely, with appropriate tables and figures and statistical analysis.

It will be more beneficial to include additional representative images of histopathological findings to enhance the visual understanding of the results.

Discussions:

The discussion provides a comprehensive interpretation of the result in the context of existing knowledge.

Conclusion:

The conclusion succinctly summarizes the main findings and their implications.

It would be helpful to include suggestions for future research directions.

References:

The references are recent and up to date, adding to the currency of the paper.

The references are highly relevant to the topic matter being discussed.

The author appropriately draws connections between the references and their research.

Writing style:

The writing style is clear and concise, making it easy to follow the author's idea.

Some parts could benefit from additional organization for improved readability.

-For all figures, it would be beneficial to include micrometer measurements to assist in size estimation. Additionally, enhancing the photographs would help identify the described colorations (e.g., the blue DAPI coloration is not apparent in Fig. 1). In my humble opinion, including additional H&E staining pictures would greatly contribute to proper comparison and description of the individual tissue structures.

-In Fig. 1b, the inset is not distinguishable from the original picture.

Finally, I believe the work is worthy of publication once these minor corrections have been taken into consideration.

Reviewers' comments:

Reviewer's Responses to Questions

**Comments to the Author**

1. Is the manuscript technically sound, and do the data support the conclusions?

Reviewer #1: Yes

Reviewer #2: Yes

2. Has the statistical analysis been performed appropriately and rigorously? 

Reviewer #1: Yes

Reviewer #2: N/A

3. Have the authors made all data underlying the findings in their manuscript fully available?

Reviewer #1: Yes

Reviewer #2: Yes

4. Is the manuscript presented in an intelligible fashion and written in standard English?

Reviewer #1: Yes

Reviewer #2: Yes

5. Review Comments to the Author

Reviewer #1: Title of paper:

Histological evaluation of the distribution of systemic AA-amyloidosis in nine domestic shorthair cats

The abstract:

The abstract summarizes the obtained findings, but the histopathological findings need more focusing.

Introduction:

The objective of this study can be written in a better way and more interesting.

The Authers not highlighting the lack of information about the histopathological findings that could be the objective of this study.

In general, the introduction has sufficient information about the nature of amyloidosis and its prevalence.

Material and Methods:

Materials and methods were written subtitled in a good way but why not talk about precautions taken during collecting testes samples?

Consider providing more details on specific staining techniques, reagents and equipment used.

Results:

The results are presented clearly and concisely, with appropriate tables and figures and statistical analysis.

It will be more beneficial to include additional representative images of histopathological findings to enhance the visual understanding of the results.

Discussions:

The discussion provides a comprehensive interpretation of the result in the context of existing knowledge.

Conclusion:

The conclusion succinctly summarizes the main findings and their implications.

It would be helpful to include suggestions for future research directions.

References:

The references are recent and up to date, adding to the currency of the paper.

The references are highly relevant to the topic matter being discussed.

The author appropriately draws connections between the references and their research.

Writing style:

The writing style is clear and concise, making it easy to follow the author's idea.

Some parts could benefit from additional organization for improved readability.

Additional Comments:

The study represents a valuable contribution to the field of hitopathology and provides novel insights into amyloidosis in cats.

Reviewer #2: Dear Sir,

Regarding your manuscript entitled "Histological evaluation of the distribution of systemic AA-amyloidosis in nine domestic shorthair cats," I would like to express my opinion. In my personal view, the work is an excellent piece of research that sheds descriptive light on AA-amyloidosis in domestic shorthair cats.

I have a few comments on the manuscript, mostly regarding the quality of its figures. I kindly request your patience as I provide my feedback:

-For all figures, it would be beneficial to include micrometer measurements to assist in size estimation. Additionally, enhancing the photographs would help identify the described colorations (e.g., the blue DAPI coloration is not apparent in Fig. 1). In my humble opinion, including additional H&E staining pictures would greatly contribute to proper comparison and description of the individual tissue structures.

-In Fig. 1b, the inset is not distinguishable from the original picture.

Finally, I believe the work is worthy of publication once these minor corrections have been taken into consideration.

Thank you very much.

Best regards,

6. PLOS authors have the option to publish the peer review history of their article (what does this mean?). If published, this will include your full peer review and any attached files.

Reviewer #1: **Yes: **Ashraf H. Soror

Reviewer #2: **Yes: **Alkhateib Y. Gaafar

---

## [Author Response · Author response to Decision Letter 0]

17 Oct 2023

Reviewer #1

Title of paper:

Histological evaluation of the distribution of systemic AA-amyloidosis in nine domestic shorthair cats

The abstract:

The abstract summarizes the obtained findings, but the histopathological findings need more focusing.

- Authors response: thank you for your comment. We added few more information on the histopathological findings.

Introduction:

The objective of this study can be written in a better way and more interesting.

- Authors response: thank you for your comment. We changed the final part of the introduction according to your comment.

The Authers not highlighting the lack of information about the histopathological findings that could be the objective of this study.

- Authors response: we also added this missing information in the introduction.

In general, the introduction has sufficient information about the nature of amyloidosis and its prevalence.

Material and Methods:

Materials and methods were written subtitled in a good way but why not talk about precautions taken during collecting testes samples?

- Authors response: thank you for this comment, we added more details on how sample collection was performed.

Consider providing more details on specific staining techniques, reagents and equipment used.

- Authors response: we added information on Congo Red staining reagent. In our opinion, the rest of the materials and methods description is detailed.

Results:

The results are presented clearly and concisely, with appropriate tables and figures and statistical analysis.

It will be more beneficial to include additional representative images of histopathological findings to enhance the visual understanding of the results.

- Authors response: thank you for your comments. We added one image of a tongue to better show the architecture of the tissue and the localization of the amyloid. In our opinion, the included pictures are already numerous and the most representative for amyloid distribution. Although, we added symbols and labels to better identify the structures and we also improved the legends with more detailed descriptions to guide the reader.

Discussions:

The discussion provides a comprehensive interpretation of the result in the context of existing knowledge.

- Authors response: thank you for your comment.

Conclusion:

The conclusion succinctly summarizes the main findings and their implications.

It would be helpful to include suggestions for future research directions.

- Authors response: thank you for these comments. We added one sentence to suggest possible further investigations.

References:

The references are recent and up to date, adding to the currency of the paper.

The references are highly relevant to the topic matter being discussed.

The author appropriately draws connections between the references and their research.

- Authors response: thank you for your comment.

Writing style:

The writing style is clear and concise, making it easy to follow the author's idea.

Some parts could benefit from additional organization for improved readability.

- Authors response: thank you for your comment. We rephrased some sentences trying to improve the readability.

Additional Comments:

The study represents a valuable contribution to the field of hitopathology and provides novel insights into amyloidosis in cats.

Reviewer #2: 

Dear Sir,

Regarding your manuscript entitled "Histological evaluation of the distribution of systemic AA-amyloidosis in nine domestic shorthair cats," I would like to express my opinion. In my personal view, the work is an excellent piece of research that sheds descriptive light on AA-amyloidosis in domestic shorthair cats.

I have a few comments on the manuscript, mostly regarding the quality of its figures. I kindly request your patience as I provide my feedback:

-For all figures, it would be beneficial to include micrometer measurements to assist in size estimation. Additionally, enhancing the photographs would help identify the described colorations (e.g., the blue DAPI coloration is not apparent in Fig. 1). In my humble opinion, including additional H&E staining pictures would greatly contribute to proper comparison and description of the individual tissue structures.

- Authors response: thank you for your comment. We added a scale bar in the pictures, included a H&E picture of the tongue, adjusted the contrast of some images and changed the plate for spleen pictures, to better represent amyloid distribution. In our opinion, new H&E images give no additional details because they are not representative of amyloid presence and distribution. Since the images are already numerous, we preferred instead to better explain the Congo Red pictures improving the legends and adding labels to help the reader. We hope you can find it useful.

-In Fig. 1b, the inset is not distinguishable from the original picture.

- Authors response: thank you for your comment. We corrected Fig. 1 b to highlight the inset.

Finally, I believe the work is worthy of publication once these minor corrections have been taken into consideration.

Thank you very much.

Best regards,

---

## [Editor Report · Decision Letter 1]

23 Oct 2023

Histological evaluation of the distribution of systemic AA-amyloidosis in nine domestic shorthair cats

PONE-D-23-16620R1

Dear Dr. Moccia

We’re pleased to inform you that your manuscript has been judged scientifically suitable for publication and will be formally accepted for publication once it meets all outstanding technical requirements.

Kind regards,

Khaled Abd EL-Hamid Abd EL-Razik, Ph.D.

Academic Editor

PLOS ONE
---

## [Editor Report · Acceptance letter]

26 Oct 2023

PONE-D-23-16620R1 

Histological evaluation of the distribution of systemic AA-amyloidosis in nine domestic shorthair cats 

Dear Dr. Moccia:

I'm pleased to inform you that your manuscript has been deemed suitable for publication in PLOS ONE. Congratulations! Your manuscript is now with our production department. 

Kind regards, 

on behalf of

Dr. Khaled Abd EL-Hamid Abd EL-Razik 

Academic Editor

PLOS ONE